# A Locally Adaptive Normal Distribution

**Georgios Arvanitidis, Lars Kai Hansen and Søren Hauberg**
Technical University of Denmark, Lyngby, Denmark
DTU Compute, Section for Cognitive Systems
{gear,lkai,sohau}@dtu.dk

## Abstract

The multivariate normal density is a monotonic function of the distance to the mean, and its ellipsoidal shape is due to the underlying Euclidean metric. We suggest to replace this metric with a locally adaptive, smoothly changing (Riemannian) metric that favors regions of high local density. The resulting *locally adaptive normal distribution (LAND)* is a generalization of the normal distribution to the "manifold" setting, where data is assumed to lie near a potentially low-dimensional manifold embedded in $\mathbb{R}^D$. The LAND is parametric, depending only on a mean and a covariance, and is the maximum entropy distribution under the given metric. The underlying metric is, however, non-parametric. We develop a maximum likelihood algorithm to infer the distribution parameters that relies on a combination of gradient descent and Monte Carlo integration. We further extend the LAND to mixture models, and provide the corresponding EM algorithm. We demonstrate the efficiency of the LAND to fit non-trivial probability distributions over both synthetic data, and EEG measurements of human sleep.

## 1 Introduction

The multivariate normal distribution is a fundamental building block in many machine learning algorithms, and its well-known density can compactly be written as

$$p(\mathbf{x} \mid \boldsymbol{\mu}, \boldsymbol{\Sigma}) \propto \exp\left(-\frac{1}{2}\mathrm{dist}_{\boldsymbol{\Sigma}}^2(\boldsymbol{\mu}, \mathbf{x})\right), \tag{1}$$

where $\mathrm{dist}_{\boldsymbol{\Sigma}}^2(\boldsymbol{\mu}, \mathbf{x})$ denotes the Mahalanobis distance for covariance matrix $\boldsymbol{\Sigma}$. This distance measure corresponds to the length of the straight line connecting $\boldsymbol{\mu}$ and $\mathbf{x}$, and consequently the normal distribution is often used to model *linear* phenomena. When data lies near a nonlinear manifold embedded in $\mathbb{R}^D$ the normal distribution becomes inadequate due to its linear metric. We investigate if a useful distribution can be constructed by replacing the linear distance function with a nonlinear counterpart. This is similar in spirit to Isomap [21] that famously replace the linear distance with a geodesic distance measured over a neighborhood graph spanned by the data, thereby allowing for a nonlinear model. This is, however, a discrete distance measure that is only well-defined over the training data. For a generative model, we need a continuously defined metric over the entire $\mathbb{R}^D$.

Following Hauberg et al. [9] we learn a smoothly changing metric that favors regions of high density i.e., geodesics tend to move near the data. Under this metric, the data space is interpreted as a $D$-dimensional Riemannian manifold. This "manifold learning" does not change dimensionality, but merely provides a local description of the data. The Riemannian view-point, however, gives a strong mathematical foundation upon which the proposed distribution can be developed. Our work, thus, bridges work on statistics on Riemannian manifolds [15, 23] with manifold learning [21].

We develop a *locally adaptive normal distribution (LAND)* as follows: First, we construct a metric that captures the nonlinear structure of the data and enables us to compute geodesics; from this, an

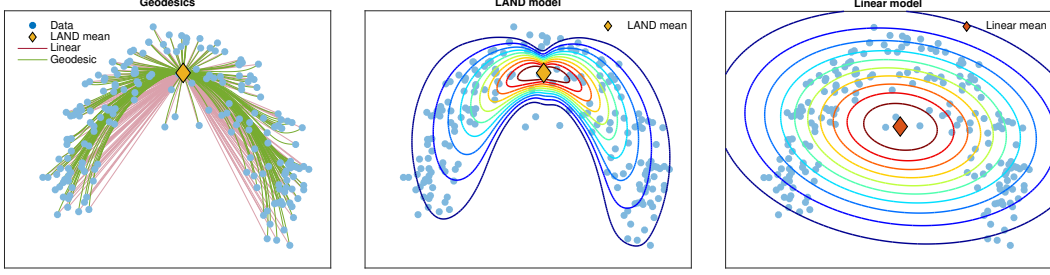

Figure 1: Illustration of the LAND using MNIST images of the digit 1 projected onto the first 2 principal components. *Left*: comparison of the geodesic and the linear distance. *Center*: the proposed locally adaptive normal distribution. *Right*: the Euclidean normal distribution.

unnormalized density is trivially defined. Second, we propose a scalable Monte Carlo integration scheme for normalizing the density with respect to the measure induced by the metric. Third, we develop a gradient-based algorithm for maximum likelihood estimation on the learned manifold. We further consider a mixture of LANDs and provide the corresponding EM algorithm. The usefulness of the model is verified on both synthetic data and EEG measurements of human sleep stages.

**Notation**: all points $\mathbf{x} \in \mathbb{R}^D$ are considered as column vectors, and they are denoted with bold lowercase characters. $\mathcal{S}_{++}^D$ represents the set of symmetric $D \times D$ positive definite matrices. The learned Riemannian manifold is denoted $\mathcal{M}$, and its tangent space at $\mathbf{x} \in \mathcal{M}$ is denoted $\mathcal{T}_{\mathbf{x}}\mathcal{M}$.

## 2 A Brief Summary of Riemannian Geometry

We start our exposition with a brief review of *Riemannian manifolds* [6]. These smooth manifolds are naturally equipped with a distance measure, and are commonly used to model physical phenomena such as dynamical or periodic systems, and many problems that have a smooth behavior.

**Definition 1.** *A smooth manifold $\mathcal{M}$ together with a Riemannian metric $\mathbf{M} : \mathcal{M} \to \mathcal{S}_{++}^D$ is called a Riemannian manifold. The Riemannian metric $\mathbf{M}$ encodes a smoothly changing inner product $\langle \mathbf{u}, \mathbf{M}(\mathbf{x})\mathbf{v} \rangle$ on the tangent space $\mathbf{u}, \mathbf{v} \in \mathcal{T}_{\mathbf{x}}\mathcal{M}$ of each point $\mathbf{x} \in \mathcal{M}$.*

**Remark 1.** *The Riemannian metric $\mathbf{M}(\mathbf{x})$ acts on tangent vectors, and may, thus, be interpreted as a standard Mahalanobis metric restricted to an infinitesimal region around $\mathbf{x}$.*

The local inner product based on $\mathbf{M}$ is a suitable model for capturing local behavior of data, i.e. *manifold learning*. From the inner product, we can define *geodesics* as length-minimizing curves connecting two points $\mathbf{x}, \mathbf{y} \in \mathcal{M}$, i.e.

$$\hat{\boldsymbol{\gamma}} = \underset{\boldsymbol{\gamma}}{\operatorname{argmin}} \int_0^1 \sqrt{\langle \boldsymbol{\gamma}'(t), \mathbf{M}(\boldsymbol{\gamma}(t))\boldsymbol{\gamma}'(t) \rangle} \mathrm{d}t, \quad \text{s.t.} \quad \boldsymbol{\gamma}(0) = \mathbf{x}, \ \boldsymbol{\gamma}(1) = \mathbf{y}. \tag{2}$$

Here $\mathbf{M}(\boldsymbol{\gamma}(t))$ is the metric tensor at $\boldsymbol{\gamma}(t)$, and the tangent vector $\boldsymbol{\gamma}'$ denotes the derivative (velocity) of $\boldsymbol{\gamma}$. The distance between $\mathbf{x}$ and $\mathbf{y}$ is defined as the length of the geodesic. A standard result from differential geometry is that the geodesic can be found as the solution to a system of $2^{\text{nd}}$ order ordinary differential equations (ODEs) [6, 9]:

$$\boldsymbol{\gamma}''(t) = -\frac{1}{2}\mathbf{M}^{-1}(\boldsymbol{\gamma}(t)) \left[ \frac{\partial \mathrm{vec}[\mathbf{M}(\boldsymbol{\gamma}(t))]}{\partial \boldsymbol{\gamma}(t)} \right]^{\mathsf{T}} (\boldsymbol{\gamma}'(t) \otimes \boldsymbol{\gamma}'(t)) \tag{3}$$

subject to $\boldsymbol{\gamma}(0) = \mathbf{x}$, $\boldsymbol{\gamma}(1) = \mathbf{y}$. Here vec[·] stacks the columns of a matrix into a vector and $\otimes$ is the Kronecker product.

This differential equation allows us to define basic operations on the manifold. The *exponential map* at a point $\mathbf{x}$ takes a tangent vector $\mathbf{v} \in \mathcal{T}_{\mathbf{x}}\mathcal{M}$ to $\mathbf{y} = \mathrm{Exp}_{\mathbf{x}}(\mathbf{v}) \in \mathcal{M}$ such that the curve $\boldsymbol{\gamma}(t) = \mathrm{Exp}_{\mathbf{x}}(t \cdot \mathbf{v})$ is a geodesic originating at $\mathbf{x}$ with initial

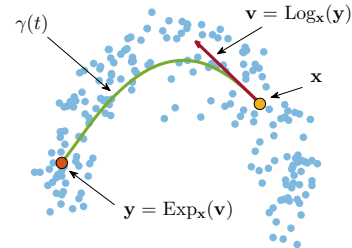

Figure 2: An illustration of the exponential and logarithmic maps.

velocity $\mathbf{v}$ and length $\|\mathbf{v}\|$. The inverse mapping, which takes $\mathbf{y}$ to $\mathcal{T}_{\mathbf{x}}\mathcal{M}$ is known as the *logarithm map* and is denoted $\mathrm{Log}_{\mathbf{x}}(\mathbf{y})$. By definition $\|\mathrm{Log}_{\mathbf{x}}(\mathbf{y})\|$ corresponds to the geodesic distance from $\mathbf{x}$ to $\mathbf{y}$. These operations are illustrated in Fig. 2. The exponential and the logarithmic map can be computed by solving Eq. 3 numerically, as an *initial value problem (IVP)* or a *boundary value problem (BVP)* respectively. In practice the IVPs are substantially faster to compute than the BVPs.

The Mahalanobis distance is naturally extended to Riemannian manifolds as $\mathrm{dist}^2_{\boldsymbol{\Sigma}}(\mathbf{x}, \mathbf{y}) = \langle \mathrm{Log}_{\mathbf{x}}(\mathbf{y}), \boldsymbol{\Sigma}^{-1}\mathrm{Log}_{\mathbf{x}}(\mathbf{y}) \rangle$. From this, Pennec [15] considered the Riemannian normal distribution

$$p_{\mathcal{M}}(\mathbf{x} \mid \boldsymbol{\mu}, \boldsymbol{\Sigma}) = \frac{1}{C} \exp\left(-\frac{1}{2}\langle \mathrm{Log}_{\boldsymbol{\mu}}(\mathbf{x}), \boldsymbol{\Sigma}^{-1}\mathrm{Log}_{\boldsymbol{\mu}}(\mathbf{x}) \rangle\right), \quad \mathbf{x} \in \mathcal{M} \tag{4}$$

and showed that it is the manifold-valued distribution with maximum entropy subject to a known mean and covariance. This distribution is an instance of Eq. 1 and is the distribution we consider in this paper. Next, we consider standard "intrinsic least squares" estimates of $\boldsymbol{\mu}$ and $\boldsymbol{\Sigma}$.

## 2.1 Intrinsic Least Squares Estimators

Let the data be generated from an unknown probability distribution $q_{\mathcal{M}}(\mathbf{x})$ on a manifold. Then it is common [15] to define the *intrinsic mean* of the distribution as the point that minimize the variance

$$\hat{\boldsymbol{\mu}} = \underset{\boldsymbol{\mu} \in \mathcal{M}}{\mathrm{argmin}} \int_{\mathcal{M}} \mathrm{dist}^2(\boldsymbol{\mu}, \mathbf{x}) q_{\mathcal{M}}(\mathbf{x}) \mathrm{d}\mathcal{M}(\mathbf{x}), \tag{5}$$

where $\mathrm{d}\mathcal{M}(\mathbf{x})$ is the measure (or infinitesimal volume element) induced by the metric. Based on the mean, a covariance matrix can be defined

$$\hat{\boldsymbol{\Sigma}} = \int_{\mathcal{D}(\hat{\boldsymbol{\mu}})} \mathrm{Log}_{\hat{\boldsymbol{\mu}}}(\mathbf{x})\mathrm{Log}_{\hat{\boldsymbol{\mu}}}(\mathbf{x})^{\intercal} q_{\mathcal{M}}(\mathbf{x})\mathrm{d}\mathcal{M}(\mathbf{x}), \tag{6}$$

where $\mathcal{D}(\hat{\boldsymbol{\mu}})$ is the domain over which $\mathcal{T}_{\hat{\boldsymbol{\mu}}}\mathcal{M}$ is well-defined. For the manifolds we consider, the domain $\mathcal{D}(\hat{\boldsymbol{\mu}})$ is $\mathbb{R}^D$. Practical estimators of $\hat{\boldsymbol{\mu}}$ rely on gradient-based optimization to find a local minimizer of Eq. 5, which is well-defined [12]. For finite data $\{\mathbf{x}_n\}_{n=1}^N$, the descent direction is proportional to $\hat{\mathbf{v}} = \sum_{n=1}^N \mathrm{Log}_{\boldsymbol{\mu}}(\mathbf{x}_n) \in \mathcal{T}_{\boldsymbol{\mu}}\mathcal{M}$, and the updated mean is a point on the geodesic curve $\boldsymbol{\gamma}(t) = \mathrm{Exp}_{\boldsymbol{\mu}}(t \cdot \hat{\mathbf{v}})$. After estimating the mean, the empirical covariance matrix is estimated as $\hat{\boldsymbol{\Sigma}} = \frac{1}{N-1}\sum_{n=1}^N \mathrm{Log}_{\hat{\boldsymbol{\mu}}}(\mathbf{x}_n)\mathrm{Log}_{\hat{\boldsymbol{\mu}}}(\mathbf{x}_n)^{\intercal}$. It is worth noting that even though these estimators are natural, they are not maximum likelihood estimates for the Riemannian normal distribution (4).

In practice, the intrinsic mean often falls in regions of low data density [8]. For instance, consider data distributed uniformly on the equator of a sphere, then the optima of Eq. 5 is either of the poles. Consequently, the empirical covariance is often overestimated.

## 3 A Locally Adaptive Normal Distribution

We now have the tools to define a locally adaptive normal distribution (LAND): we replace the linear Euclidean distance with a locally adaptive Riemannian distance and study the corresponding Riemannian normal distribution (4). By learning a Riemannian manifold and using its structure to estimate distributions of the data, we provide a new and useful link between Riemannian statistics and manifold learning.

### 3.1 Constructing a Metric

In the context of manifold learning, Hauberg et al. [9] suggest to model the local behavior of the data manifold via a locally-defined Riemannian metric. Here we propose to use a local covariance matrix to represent the local structure of the data. We only consider diagonal covariances for computational efficiency and to prevent the overfitting. The locality of the covariance is defined via an isotropic Gaussian kernel of size $\sigma$. Thus, the metric tensor at $\mathbf{x} \in \mathcal{M}$ is defined as the inverse of a local diagonal covariance matrix with entries

$$M_{dd}(\mathbf{x}) = \left(\sum_{n=1}^N w_n(\mathbf{x})(x_{nd} - x_d)^2 + \rho\right)^{-1}, \quad \text{with} \quad w_n(\mathbf{x}) = \exp\left(-\frac{\|\mathbf{x}_n - \mathbf{x}\|_2^2}{2\sigma^2}\right). \tag{7}$$

Here $x_{nd}$ is the $d^{\text{th}}$ dimension of the $n^{\text{th}}$ observation, and $\rho$ a regularization parameter to avoid singular covariances. This defines a smoothly changing (hence Riemannian) metric that captures the local structure of the data. It is easy to see that if $\mathbf{x}$ is outside of the support of the data, then the metric tensor is large. Thus, geodesics are "pulled" towards the data where the metric is small. Note that the proposed metric is not invariant to linear transformations.While we restrict our attention to this particular choice, other learned metrics are equally applicable, c.f. [22, 9].

## 3.2   Estimating the Normalization Constant

The normalization constant of Eq. 4 is by definition

$$\mathcal{C}(\boldsymbol{\mu}, \boldsymbol{\Sigma}) = \int_{\mathcal{M}} \exp\left(-\frac{1}{2}\langle \text{Log}_{\boldsymbol{\mu}}(\mathbf{x}), \boldsymbol{\Sigma}^{-1}\text{Log}_{\boldsymbol{\mu}}(\mathbf{x})\rangle\right) d\mathcal{M}(\mathbf{x}), \tag{8}$$

where $d\mathcal{M}(\mathbf{x})$ denotes the measure induced by the Riemannian metric. The constant $\mathcal{C}(\boldsymbol{\mu}, \boldsymbol{\Sigma})$ depends not only on the covariance matrix, but also on the mean of the distribution, and the curvature of the manifold (captured by the logarithm map). For a general learned manifold, $\mathcal{C}(\boldsymbol{\mu}, \boldsymbol{\Sigma})$ is inaccessible in closed-form and we resort to numerical techniques. We start by rewriting Eq. 8 as

$$\mathcal{C}(\boldsymbol{\mu}, \boldsymbol{\Sigma}) = \int_{\mathcal{T}_{\boldsymbol{\mu}}\mathcal{M}} \sqrt{\left|\mathbf{M}(\text{Exp}_{\boldsymbol{\mu}}(\mathbf{v}))\right|} \exp\left(-\frac{1}{2}\langle \mathbf{v}, \boldsymbol{\Sigma}^{-1}\mathbf{v}\rangle\right) d\mathbf{v}. \tag{9}$$

In effect, we integrate the distribution over the tangent space $\mathcal{T}_{\boldsymbol{\mu}}\mathcal{M}$ instead of directly over the manifold. This transformation relies on the fact that the volume of an infinitely small area on the manifold can be computed in the tangent space if we take the deformation of the metric into account [15]. This deformation is captured by the measure which, in the tangent space, is $d\mathcal{M}(\mathbf{x}) = \sqrt{\left|\mathbf{M}(\text{Exp}_{\boldsymbol{\mu}}(\mathbf{v}))\right|}d\mathbf{v}$. For notational simplicity we define the function $m(\boldsymbol{\mu}, \mathbf{v}) = \sqrt{\left|\mathbf{M}(\text{Exp}_{\boldsymbol{\mu}}(\mathbf{v}))\right|}$, which intuitively captures the cost for a point to be outside the data support ($m$ is large in low density areas and small where the density is high).

We estimate the normalization constant (9) using Monte Carlo integration. We first multiply and divide the integral with the normalization constant of the Euclidean normal distribution $\mathcal{Z} = \sqrt{(2\pi)^D |\boldsymbol{\Sigma}|}$. Then, the integral becomes an expectation estimation problem $\mathcal{C}(\boldsymbol{\mu}, \boldsymbol{\Sigma}) = \mathcal{Z} \cdot \mathbb{E}_{\mathcal{N}(0,\boldsymbol{\Sigma})}[m(\boldsymbol{\mu}, \mathbf{v})]$, which can be estimated numerically as

$$\mathcal{C}(\boldsymbol{\mu}, \boldsymbol{\Sigma}) \simeq \frac{\mathcal{Z}}{S}\sum_{s=1}^{S} m(\boldsymbol{\mu}, \mathbf{v}_s), \quad \text{where} \quad \mathbf{v}_s \sim \mathcal{N}(0, \boldsymbol{\Sigma}) \tag{10}$$

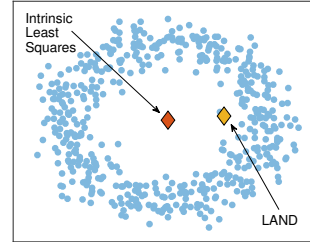

Figure 3:   Comparison of LAND and intrinsic least squares means.

and $S$ is the number of samples on $\mathcal{T}_{\boldsymbol{\mu}}\mathcal{M}$. The computationally expensive element is to evaluate $m$, which in turn requires evaluating $\text{Exp}_{\boldsymbol{\mu}}(\mathbf{v})$. This amounts to solving an IVP numerically, which is fairly fast. Had we performed the integration directly on the manifold (8) we would have had to evaluate the logarithm map, which is a much more expensive BVP. The tangent space integration, thus, scales better.

## 3.3   Inferring Parameters

Assuming an independent and identically distributed dataset $\{\mathbf{x}_n\}_{n=1}^{N}$, we can write their joint distribution as $p_{\mathcal{M}}(\mathbf{x}_1, \ldots, \mathbf{x}_N) = \prod_{n=1}^{N} p_{\mathcal{M}}(\mathbf{x}_n \mid \boldsymbol{\mu}, \boldsymbol{\Sigma})$. We find parameters $\boldsymbol{\mu}$ and $\boldsymbol{\Sigma}$ by maximum likelihood, which we implement by minimizing the mean negative log-likelihood

$$\{\hat{\boldsymbol{\mu}}, \hat{\boldsymbol{\Sigma}}\} = \underset{\substack{\boldsymbol{\mu}\in\mathcal{M} \\ \boldsymbol{\Sigma}\in\mathcal{S}_{++}^{D}}}{\text{argmin}} \phi\left(\boldsymbol{\mu}, \boldsymbol{\Sigma}\right) = \underset{\substack{\boldsymbol{\mu}\in\mathcal{M} \\ \boldsymbol{\Sigma}\in\mathcal{S}_{++}^{D}}}{\text{argmin}} \frac{1}{2N}\sum_{n=1}^{N}\langle \text{Log}_{\boldsymbol{\mu}}(\mathbf{x}_n), \boldsymbol{\Sigma}^{-1}\text{Log}_{\boldsymbol{\mu}}(\mathbf{x}_n)\rangle + \log\left(\mathcal{C}(\boldsymbol{\mu}, \boldsymbol{\Sigma})\right). \tag{11}$$

The first term of the objective function $\phi : \mathcal{M} \times \mathcal{S}_{++}^{D}$ is a data-fitting term, while the second can be seen as a force that both pulls the mean closer to the high density areas and shrinks the covariance. Specifically, when the mean is in low density areas, as well as when the covariance gives significant

probability to those areas, the value of $m(\boldsymbol{\mu}, \mathbf{v})$ will by construction be large. Consequently, $\mathcal{C}(\boldsymbol{\mu}, \boldsymbol{\Sigma})$ will increase and these solutions will be penalized. In practice, we find that the maximum likelihood LAND mean generally avoids low density regions, which is in contrast to the standard intrinsic least squares mean (5), see Fig. 3.

In practice we optimize $\phi$ using block coordinate descent: we optimize the mean keeping the covariance fixed and vice versa. Unfortunately, both of the sub-problems are non-convex, and unlike the linear normal distribution, they lack a closed-form solution. Since the logarithm map is a differentiable function, we can use gradient-based techniques to infer $\boldsymbol{\mu}$ and $\boldsymbol{\Sigma}$. Below we give the descent direction for $\boldsymbol{\mu}$ and $\boldsymbol{\Sigma}$ and the corresponding optimization scheme is given in Algorithm 1. Initialization is discussed in the supplements.

---

**Algorithm 1** LAND maximum likelihood

**Input:** the data $\{\mathbf{x}_n\}_{n=1}^N$, stepsizes $\alpha_{\boldsymbol{\mu}}, \alpha_{\mathbf{A}}$
**Output:** the estimated $\hat{\boldsymbol{\mu}}$, $\hat{\boldsymbol{\Sigma}}$, $\hat{\mathcal{C}}(\hat{\boldsymbol{\mu}}, \hat{\boldsymbol{\Sigma}})$
1: initialize $\boldsymbol{\mu}^0, \boldsymbol{\Sigma}^0$ and $t \leftarrow 0$
2: **repeat**
3:   estimate $\mathcal{C}(\boldsymbol{\mu}^t, \boldsymbol{\Sigma}^t)$ using Eq. 10
4:   compute $d_{\boldsymbol{\mu}}\phi(\boldsymbol{\mu}^t, \boldsymbol{\Sigma}^t)$ using Eq. 12
5:   $\boldsymbol{\mu}^{t+1} \leftarrow \mathrm{Exp}_{\boldsymbol{\mu}^t}(\alpha_{\boldsymbol{\mu}} d_{\boldsymbol{\mu}}\phi(\boldsymbol{\mu}^t, \boldsymbol{\Sigma}^t))$
6:   estimate $\mathcal{C}(\boldsymbol{\mu}^{t+1}, \boldsymbol{\Sigma}^t)$ using Eq. 10
7:   compute $\nabla_{\mathbf{A}}\phi(\boldsymbol{\mu}^{t+1}, \boldsymbol{\Sigma}^t)$ using Eq. 13
8:   $\mathbf{A}^{t+1} \leftarrow \mathbf{A}^t - \alpha_{\mathbf{A}}\nabla_{\mathbf{A}}\phi(\boldsymbol{\mu}^{t+1}, \boldsymbol{\Sigma}^t)$
9:   $\boldsymbol{\Sigma}^{t+1} \leftarrow [(\mathbf{A}^{t+1})^{\intercal}\mathbf{A}^{t+1}]^{-1}$
10:   $t \leftarrow t + 1$
11: **until** $\left\| \phi(\boldsymbol{\mu}^{t+1}, \boldsymbol{\Sigma}^{t+1}) - \phi(\boldsymbol{\mu}^t, \boldsymbol{\Sigma}^t) \right\|_2^2 \leq \epsilon$

---

**Optimizing $\boldsymbol{\mu}$**: the objective function is differentiable with respect to $\boldsymbol{\mu}$ [6], and using that $\frac{\partial}{\partial \boldsymbol{\mu}}\langle \mathrm{Log}_{\boldsymbol{\mu}}(\mathbf{x}), \boldsymbol{\Sigma}^{-1}\mathrm{Log}_{\boldsymbol{\mu}}(\mathbf{x})\rangle = -2\boldsymbol{\Sigma}^{-1}\mathrm{Log}_{\boldsymbol{\mu}}(\mathbf{x})$, we get the gradient

$$\nabla_{\boldsymbol{\mu}}\phi(\boldsymbol{\mu}, \boldsymbol{\Sigma}) = -\boldsymbol{\Sigma}^{-1}\left[ \frac{1}{N}\sum_{n=1}^N \mathrm{Log}_{\boldsymbol{\mu}}(\mathbf{x}_n) - \frac{\mathcal{Z}}{\mathcal{C}(\boldsymbol{\mu}, \boldsymbol{\Sigma}) \cdot S}\sum_{s=1}^S m(\boldsymbol{\mu}, \mathbf{v}_s)\mathbf{v}_s \right]. \tag{12}$$

It is easy to see that this gradient is highly dependent on the condition number of $\boldsymbol{\Sigma}$. We find that this, at times, makes the gradient unstable, and choose to use the steepest descent direction instead of the gradient direction. This is equal to $d_{\boldsymbol{\mu}}\phi(\boldsymbol{\mu}, \boldsymbol{\Sigma}) = -\boldsymbol{\Sigma}\nabla_{\boldsymbol{\mu}}\phi(\boldsymbol{\mu}, \boldsymbol{\Sigma})$ (see supplements).

**Optimizing $\boldsymbol{\Sigma}$**: since the covariance matrix by definition is constrained to be in the space $\mathcal{S}_{++}^D$, a common trick is to decompose the matrix as $\boldsymbol{\Sigma}^{-1} = \mathbf{A}^{\intercal}\mathbf{A}$, and optimize the objective with respect to $\mathbf{A}$. The gradient of this factor is (see supplements for derivation)

$$\nabla_{\mathbf{A}}\phi(\boldsymbol{\mu}, \boldsymbol{\Sigma}) = \mathbf{A}\left[ \frac{1}{N}\sum_{n=1}^N \mathrm{Log}_{\boldsymbol{\mu}}(\mathbf{x}_n)\mathrm{Log}_{\boldsymbol{\mu}}(\mathbf{x}_n)^{\intercal} - \frac{\mathcal{Z}}{\mathcal{C}(\boldsymbol{\mu}, \boldsymbol{\Sigma}) \cdot S}\sum_{s=1}^S m(\boldsymbol{\mu}, \mathbf{v}_s)\mathbf{v}_s\mathbf{v}_s^{\intercal} \right]. \tag{13}$$

Here the first term fits the given data by increasing the size of the covariance matrix, while the second term regularizes the covariance towards a small matrix.

### 3.4 Mixture of LANDs

At this point we can find maximum likelihood estimates of the LAND model. We can easily extend this to mixtures of LANDs: Following the derivation of the standard Gaussian mixture model [3], our objective function for inferring the parameters of the LAND mixture model is formulated as follows

$$\psi(\boldsymbol{\Theta}) = \sum_{k=1}^K \sum_{n=1}^N r_{nk}\left[ \frac{1}{2}\langle \mathrm{Log}_{\boldsymbol{\mu}_k}(\mathbf{x}_n), \boldsymbol{\Sigma}_k^{-1}\mathrm{Log}_{\boldsymbol{\mu}_k}(\mathbf{x}_n)\rangle + \log(\mathcal{C}(\boldsymbol{\mu}_k, \boldsymbol{\Sigma}_k)) - \log(\pi_k) \right], \tag{14}$$

where $\boldsymbol{\Theta} = \{\boldsymbol{\mu}_k, \boldsymbol{\Sigma}_k\}_{k=1}^K$, $r_{nk} = \frac{\pi_k p_{\mathcal{M}}(\mathbf{x}_n \mid \boldsymbol{\mu}_k, \boldsymbol{\Sigma}_k)}{\sum_{l=1}^K \pi_l p_{\mathcal{M}}(\mathbf{x}_n \mid \boldsymbol{\mu}_l, \boldsymbol{\Sigma}_l)}$ is the probability that $\mathbf{x}_n$ is generated by the $k^{\mathrm{th}}$ component, and $\sum_{k=1}^K \pi_k = 1$, $\pi_k \geq 0$. The corresponding EM algorithm is in the supplements.

## 4 Experiments

In this section we present both synthetic and real experiments to demonstrate the advantages of the LAND. We compare our model with both the Gaussian mixture model (GMM), and a mixture of LANDs using least squares (LS) estimators (5, 6). Since the latter are not maximum likelihood estimates we use a Riemannian $K$-means algorithm to find cluster centers. In all experiments we use $S = 3000$ samples in the Monte Carlo integration. This choice is investigated empirically in the supplements. Furthermore, we choose $\sigma$ as small as possible, while ensuring that the manifold is smooth enough that geodesics can be computed numerically.

## 4.1 Synthetic Data Experiments

As a first experiment, we generate a nonlinear data-manifold by sampling from a mixture of 20 Gaussians positioned along a half-ellipsoidal curve (see left panel of Fig. 5). We generate 10 datasets with 300 points each, and fit for each dataset the three models with $K = 1, \ldots, 4$ number of components. Then, we generate 10000 samples from each fitted model, and we compute the mean negative log-likelihood of the true generative distribution using these samples. Fig. 4 shows that the LAND learns faster the underlying true distribution, than the GMM. Moreover, the LAND perform better than the least squares estimators, which overestimates the covariance. In the supplements we show, using the standard AIC and BIC criteria, that the optimal LAND is achieved for $K = 1$, while for the least squares estimators and the GMM, the optimal is achieved for $K = 3$ and $K = 4$ respectively.

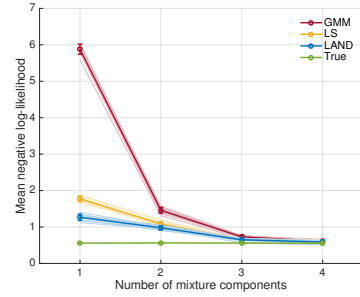

Figure 4: The mean negative log-likelihood experiment.

In addition, in Fig. 5 we show the contours for the LAND and the GMM for $K = 2$. There, we can observe that indeed, the LAND adapts locally to the data and reveals their underlying nonlinear structure. This is particularly evident near the "boundaries" of the data-manifold.

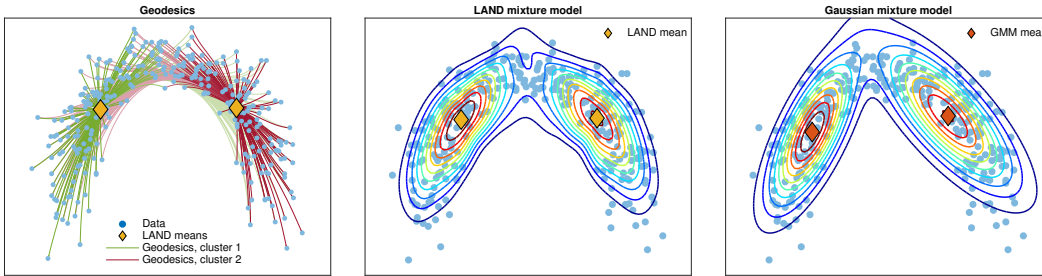

Figure 5: Synthetic data and the fitted models. *Left*: the given data, the intensity of the geodesics represent the responsibility of the point to the corresponding cluster. *Center*: the contours of the LAND mixture model. *Right*: the contours of the Gaussian mixture model.

We extend this experiment to a clustering task (see left panel of Fig. 6 for data). The center and right panels of Fig. 6 show the contours of the LAND and Gaussian mixtures, and it is evident that the LAND is substantially better at capturing non-ellipsoidal clusters. Due to space limitations, we move further illustrative experiments to the supplementary material and continue with real data.

## 4.2 Modeling Sleep Stages

We consider electro-encephalography (EEG) measurements of human sleep from 10 subjects, part of the PhysioNet database [11, 7, 5]. For each subject we get EEG measurements during sleep from two electrodes on the front and the back of the head, respectively. Measurements are sampled at $f_s = 100$Hz, and for each 30 second window a so-called sleep stage label is assigned from the set $\{1, 2, 3, 4, \text{REM}, \text{awake}\}$. Rapid eye movement (REM) sleep is particularly interesting, characterized by having EEG patterns similar to the awake state but with a complex physiological pattern, involving e.g., reduced muscle tone, rolling eye movements and erection [16]. Recent evidence points to the importance of REM sleep for memory consolidation [4]. Periods in which the sleeper is awake are typically happening in or near REM intervals. Thus we here consider the characterization of sleep in terms of three categories REM, awake, and non-REM, the latter a merger of sleep stages $1 - 4$.

We extract features from EEG measurements as follows: for each subject we subdivide the 30 second windows to 10 seconds, and apply a short-time-Fourier-transform to the EEG signal of the frontal electrode with $50\%$ overlapping windows. From this we compute the log magnitude of the spectrum $\log(1 + |f|)$ of each window. The resulting data matrix is decomposed using Non-Negative Matrix Factorization (10 random starts) into five factors, and we use the coefficients as $5D$ features. In Fig. 7 we illustrate the nonlinear manifold structure based on a three factor analysis.

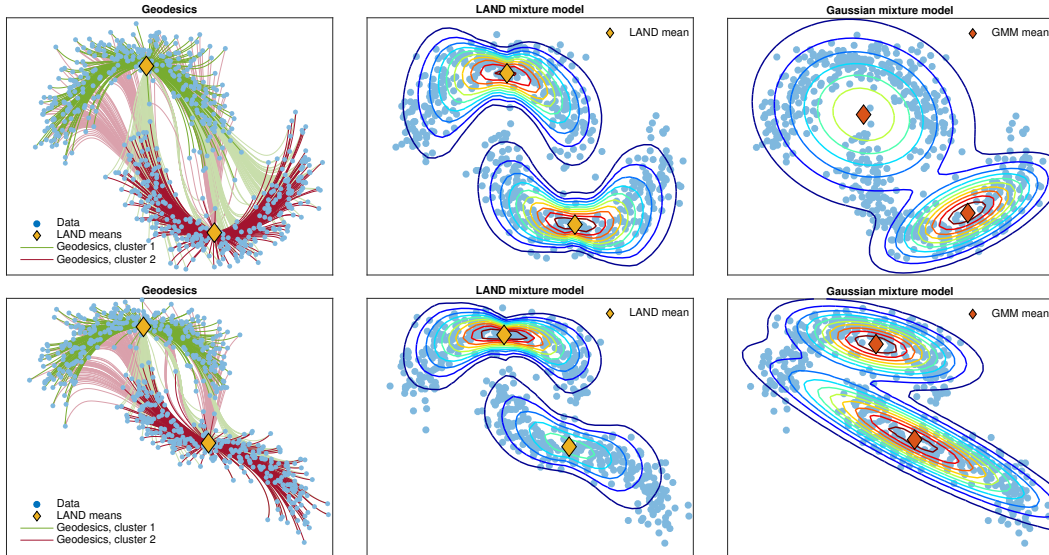

Figure 6: The clustering problem for two synthetic datasets. *Left*: the given data, the intensity of the geodesics represent the responsibility of the point to the corresponding cluster. *Center*: the LAND mixture model. *Right*: the Gaussian mixture model.

We perform clustering on the data and evaluate the alignment between cluster labels and sleep stages using the F-measure [14]. The LAND depends on the parameter $\sigma$ to construct the metric tensor, and in this experiment it is less straightforward to select $\sigma$ because of significant intersubject variability. First, we fixed $\sigma = 1$ for all the subjects. From the results in Table 1 we observe that for $\sigma = 1$ the LAND(1) generally outperforms the GMM and achieves much better alignment. To further illustrate the effect of $\sigma$ we fitted a LAND for $\sigma = [0.5, 0.6, \ldots, 1.5]$ and present the best result achieved by the LAND. Selecting $\sigma$ this way leads indeed to higher degrees of alignment further underlining that the conspicuous manifold structure and the rather compact sleep stage distributions in Fig. 7 are both captured better with the LAND representation than with a linear GMM.

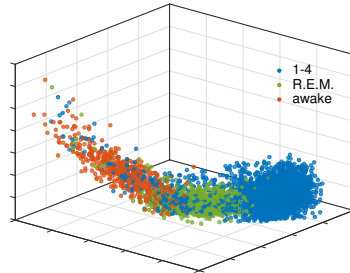

Figure 7: The 3 leading factors for subject "s151".

Table 1: The F-measure result for 10 subjects (the closer to 1 the better).

|         | s001  | s011  | s042  | s062  | s081  | s141  | s151  | s161  | s162  | s191  |
|---------|-------|-------|-------|-------|-------|-------|-------|-------|-------|-------|
| LAND(1) | **0.831** | **0.701** | 0.670 | **0.740** | **0.804** | 0.870 | **0.820** | **0.780** | 0.747 | **0.786** |
| GMM     | 0.812 | 0.690 | **0.675** | 0.651 | 0.798 | 0.870 | 0.794 | 0.775 | 0.747 | 0.776 |
| LAND    | **0.831** | **0.716** | **0.695** | **0.740** | **0.818** | **0.874** | **0.830** | **0.783** | **0.750** | **0.787** |

## 5 Related Work

We are not the first to consider Riemannian normal distributions, e.g. Pennec [15] gives a theoretical analysis of the distribution, and Zhang and Fletcher [23] consider the Riemannian counterpart of probabilistic PCA. Both consider the scenario where the manifold is known a priori. We adapt the distribution to the "manifold learning" setting by constructing a Riemannian metric that adapts to the data. This is our overarching contribution.

Traditionally, manifold learning is seen as an *embedding problem* where a low-dimensional representation of the data is sought. This is useful for visualization [21, 17, 18, 1], clustering [13], semi-supervised learning [2] and more. However, in embedding approaches, the relation between a

new point and the embedded points are less well-defined, and consequently these approaches are less suited for building generative models. In contrast, the Riemannian approach gives the ability to measure continuous geodesics that follow the structure of the data. This makes the learned Riemannian manifold a suitable space for a generative model.

Simo-Serra et al. [19] consider mixtures of Riemannian normal distributions on manifolds that are known a priori. Structurally, their EM algorithm is similar to ours, but they do not account for the normalization constants for different mixture components. Consequently, their approach is inconsistent with the probabilistic formulation. Straub et al. [20] consider data on spherical manifolds, and further consider a Dirichlet process prior for determining the number of components. Such a prior could also be incorporated in our model. The key difference to our work is that we consider learned manifolds as well as the following complications.

## 6  Discussion

In this paper we have introduced a parametric locally adaptive normal distribution. The idea is to replace the Euclidean distance in the ordinary normal distribution with a locally adaptive nonlinear distance measure. In principle, we learn a non-parametric metric space, by constructing a smoothly changing metric that induces a Riemannian manifold, where we build our model. As such, we propose a parametric model over a non-parametric space.

The non-parametric space is constructed using a local metric that is the inverse of a local covariance matrix. Here locality is defined via a Gaussian kernel, such that the manifold learning can be seen as a form of kernel smoothing. This indicates that our scheme for learning a manifold might not scale to high-dimensional input spaces. In these cases it may be more practical to learn the manifold probabilistically [22] or as a mixture of metrics [9]. This is feasible as the LAND estimation procedure is agnostic to the details of the learned manifold as long as exponential and logarithm maps can be evaluated.

Once a manifold is learned, the LAND is simply a Riemannian normal distribution. This is a natural model, but more intriguing, it is a theoretical interesting model since it is the maximum entropy distribution for a fixed mean and covariance [15]. It is generally difficult to build locally adaptive distributions with maximum entropy properties, yet the LAND does this in a fairly straight-forward manner. This is, however, only a partial truth as the distribution depends on the non-parametric space. The natural question, to which we currently do not have an answer, is whether a suitable maximum entropy manifold exist?

Algorithmically, we have proposed a maximum likelihood estimation scheme for the LAND. This combines a gradient-based optimization with a scalable Monte Carlo integration method. Once exponential and logarithm maps are available, this procedure is surprisingly simple to implement. We have demonstrated the algorithm on both real and synthetic data and results are encouraging. We almost always improve upon a standard Gaussian mixture model as the LAND is better at capturing the local properties of the data.

We note that both the manifold learning aspect and the algorithmic aspect of our work can be improved. It would be of great value to learn the parameter $\sigma$ used for smoothing the Riemannian metric, and in general, more adaptive learning schemes are of interest. Computationally, the bottleneck of our work is evaluating the logarithm maps. This may be improved by specialized solvers, e.g. probabilistic solvers [10], or manifold-specific heuristics.

The ordinary normal distribution is a key element in many machine learning algorithms. We expect that many fundamental generative models can be extended to the "manifold" setting simply by replacing the normal distribution with a LAND. Examples of this idea include Naïve Bayes, Linear Discriminant Analysis, Principal Component Analysis and more. Finally we note that standard hypothesis tests also extend to Riemannian normal distributions [15] and hence also to the LAND.

**Acknowledgements**. LKH was funded in part by the Novo Nordisk Foundation Interdisciplinary Synergy Program 2014, 'Biophysically adjusted state-informed cortex stimulation (BASICS)'. SH was funded in part by the Danish Council for Independent Research, Natural Sciences.

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
