[Supplementary Material]

# A Locally Adaptive Normal Distribution
# — Supplementary material —

**Georgios Arvanitidis, Lars Kai Hansen and Søren Hauberg**
Technical University of Denmark, Lyngby, Denmark
DTU Compute, Section for Cognitive Systems
{gear,lkai,sohau}@dtu.dk

**Notation**: all points $\mathbf{x} \in \mathbb{R}^D$ are considered as column vectors, and they are denoted with bold lowercase characters. $\mathcal{S}_{++}^D$ represents the set of symmetric $D \times D$ positive definite matrices. The learned Riemannian manifold is denoted $\mathcal{M}$, and its tangent space at point $\mathbf{x} \in \mathcal{M}$ is denoted $\mathcal{T}_{\boldsymbol{\mu}}\mathcal{M}$.

We present for convenience the domain and co-domain of the following often used terms. Note that $\mathcal{T}_{\boldsymbol{\mu}}\mathcal{M}$ is $\mathbb{R}^D$.

$$\boldsymbol{\gamma}(t) : [0,1] \to \mathcal{M} \qquad \mathrm{Exp}_{\boldsymbol{\mu}}(\mathbf{v}) : \mathcal{M} \times \mathcal{T}_{\mathbf{x}}\mathcal{M} \to \mathcal{M}$$

$$\mathbf{M}(\mathbf{x}) : \mathcal{M} \to \mathcal{S}_{++}^D \qquad \mathrm{Log}_{\boldsymbol{\mu}}(\mathbf{x}) : \mathcal{M} \times \mathcal{M} \to \mathcal{T}_{\mathbf{x}}\mathcal{M}$$

## 1 Estimating the Normalization Constant

The locally adaptive normal distribution is defined as

$$p_{\mathcal{M}}(\mathbf{x} \mid \boldsymbol{\mu}, \boldsymbol{\Sigma}) = \frac{1}{\mathcal{C}(\boldsymbol{\mu}, \boldsymbol{\Sigma})} \exp\left(-\frac{1}{2}\langle \mathrm{Log}_{\boldsymbol{\mu}}(\mathbf{x}), \boldsymbol{\Sigma}^{-1}\mathrm{Log}_{\boldsymbol{\mu}}(\mathbf{x})\rangle\right), \quad \mathbf{x} \in \mathcal{M}. \tag{15}$$

Therefore, the normalization constant is equal to

$$\int_{\mathcal{M}} p_{\mathcal{M}}(\mathbf{x} \mid \boldsymbol{\mu}, \boldsymbol{\Sigma}) \mathrm{d}\mathcal{M}(\mathbf{x}) = 1 \Rightarrow \tag{16}$$

$$\int_{\mathcal{M}} \frac{1}{\mathcal{C}(\boldsymbol{\mu}, \boldsymbol{\Sigma})} \exp\left(-\frac{1}{2}\langle \mathrm{Log}_{\boldsymbol{\mu}}(\mathbf{x}), \boldsymbol{\Sigma}^{-1}\mathrm{Log}_{\boldsymbol{\mu}}(\mathbf{x})\rangle\right) \mathrm{d}\mathcal{M}(\mathbf{x}) = 1 \Rightarrow \tag{17}$$

$$\mathcal{C}(\boldsymbol{\mu}, \boldsymbol{\Sigma}) = \int_{\mathcal{M}} \exp\left(-\frac{1}{2}\langle \mathrm{Log}_{\boldsymbol{\mu}}(\mathbf{x}), \boldsymbol{\Sigma}^{-1}\mathrm{Log}_{\boldsymbol{\mu}}(\mathbf{x})\rangle\right) \mathrm{d}\mathcal{M}(\mathbf{x}) \tag{18}$$

$$= \int_{\mathcal{D}(\boldsymbol{\mu})} \sqrt{\left|\mathbf{M}(\mathrm{Exp}_{\boldsymbol{\mu}}(\mathbf{v}))\right|} \exp\left(-\frac{1}{2}\langle \mathrm{Log}_{\boldsymbol{\mu}}(\mathrm{Exp}_{\boldsymbol{\mu}}(\mathbf{v})), \boldsymbol{\Sigma}^{-1}\mathrm{Log}_{\boldsymbol{\mu}}(\mathrm{Exp}_{\boldsymbol{\mu}}(\mathbf{v}))\rangle\right) \mathrm{d}\mathbf{v} \tag{19}$$

$$= \int_{\mathcal{T}_{\boldsymbol{\mu}}\mathcal{M}} m(\boldsymbol{\mu}, \mathbf{v}) \exp\left(-\frac{1}{2}\langle \mathbf{v}, \boldsymbol{\Sigma}^{-1}\mathbf{v}\rangle\right) \mathrm{d}\mathbf{v} \tag{20}$$

$$= \int_{\mathcal{T}_{\boldsymbol{\mu}}\mathcal{M}} m(\boldsymbol{\mu}, \mathbf{v}) \frac{\mathcal{Z}}{\mathcal{Z}} \exp\left(-\frac{1}{2}\langle \mathbf{v}, \boldsymbol{\Sigma}^{-1}\mathbf{v}\rangle\right) \mathrm{d}\mathbf{v} \tag{21}$$

$$= \mathcal{Z} \cdot \mathbb{E}_{\mathcal{N}(0,\boldsymbol{\Sigma})}[m(\boldsymbol{\mu}, \mathbf{v})] \simeq \frac{\mathcal{Z}}{S} \sum_{s=1}^{S} m(\boldsymbol{\mu}, \mathbf{v}_s), \qquad \text{where} \quad \mathbf{v}_s \sim \mathcal{N}(0, \boldsymbol{\Sigma}). \tag{22}$$

To simplify notation we have defined the $m(\boldsymbol{\mu}, \mathbf{v}) = \sqrt{\left|\mathbf{M}(\mathrm{Exp}_{\boldsymbol{\mu}}(\mathbf{v}))\right|}$ and $\mathcal{Z} = \sqrt{(2\pi)^D |\boldsymbol{\Sigma}|}$. The integral is then estimated with a Monte-Carlo technique.

## 2 Steepest Descent Direction for the Mean

The objective function is differentiable with respect to $\boldsymbol{\mu}$ with

$$\frac{\partial}{\partial\boldsymbol{\mu}}\langle\mathrm{Log}_{\boldsymbol{\mu}}(\mathbf{x}_n),\boldsymbol{\Sigma}^{-1}\mathrm{Log}_{\boldsymbol{\mu}}(\mathbf{x}_n)\rangle = -2\boldsymbol{\Sigma}^{-1}\mathrm{Log}_{\boldsymbol{\mu}}(\mathbf{x}_n) \tag{23}$$

Then the gradient of the objective function $\phi(\boldsymbol{\mu},\boldsymbol{\Sigma})$ is equal to

$$\nabla_{\boldsymbol{\mu}}\phi(\boldsymbol{\mu},\boldsymbol{\Sigma}) = \frac{\partial}{\partial\boldsymbol{\mu}}\left[\frac{1}{2N}\sum_{n=1}^{N}\langle\mathrm{Log}_{\boldsymbol{\mu}}(\mathbf{x}_n),\boldsymbol{\Sigma}^{-1}\mathrm{Log}_{\boldsymbol{\mu}}(\mathbf{x}_n)\rangle + \log(\mathcal{C}(\boldsymbol{\mu},\boldsymbol{\Sigma}))\right] \tag{24}$$

$$= -\frac{1}{N}\boldsymbol{\Sigma}^{-1}\sum_{n=1}^{N}\mathrm{Log}_{\boldsymbol{\mu}}(\mathbf{x}_n) + \frac{1}{\mathcal{C}(\boldsymbol{\mu},\boldsymbol{\Sigma})}\int_{\mathcal{M}}\frac{\partial}{\partial\boldsymbol{\mu}}\left[\exp\left(-\frac{1}{2}\langle\mathrm{Log}_{\boldsymbol{\mu}}(\mathbf{x}),\boldsymbol{\Sigma}^{-1}\mathrm{Log}_{\boldsymbol{\mu}}(\mathbf{x})\rangle\right)\right]\mathrm{d}\mathcal{M}(\mathbf{x}) \tag{25}$$

$$= -\frac{\boldsymbol{\Sigma}^{-1}}{N}\sum_{n=1}^{N}\mathrm{Log}_{\boldsymbol{\mu}}(\mathbf{x}_n) + \frac{\boldsymbol{\Sigma}^{-1}}{\mathcal{C}(\boldsymbol{\mu},\boldsymbol{\Sigma})}\int_{\mathcal{M}}\mathrm{Log}_{\boldsymbol{\mu}}(\mathbf{x})\exp\left(-\frac{1}{2}\langle\mathrm{Log}_{\boldsymbol{\mu}}(\mathbf{x}),\boldsymbol{\Sigma}^{-1}\mathrm{Log}_{\boldsymbol{\mu}}(\mathbf{x})\rangle\right)\mathrm{d}\mathcal{M}(\mathbf{x}) \tag{26}$$

$$= -\frac{1}{N}\boldsymbol{\Sigma}^{-1}\sum_{n=1}^{N}\mathrm{Log}_{\boldsymbol{\mu}}(\mathbf{x}_n) + \frac{\boldsymbol{\Sigma}^{-1}}{\mathcal{C}(\boldsymbol{\mu},\boldsymbol{\Sigma})}\int_{\mathcal{T}_{\boldsymbol{\mu}}\mathcal{M}}m(\boldsymbol{\mu},\mathbf{v})\mathbf{v}\exp\left(-\frac{1}{2}\langle\mathbf{v},\boldsymbol{\Sigma}^{-1}\mathbf{v}\rangle\right)\mathrm{d}\mathbf{v} \tag{27}$$

$$= -\boldsymbol{\Sigma}^{-1}\left[\frac{1}{N}\sum_{n=1}^{N}\mathrm{Log}_{\boldsymbol{\mu}}(\mathbf{x}_n) - \frac{\mathcal{Z}}{\mathcal{C}(\boldsymbol{\mu},\boldsymbol{\Sigma})\cdot S}\sum_{s=1}^{S}m(\boldsymbol{\mu},\mathbf{v}_s)\mathbf{v}_s\right]. \tag{28}$$

This gradient is highly dependent on the condition number of the covariance matrix $\boldsymbol{\Sigma}$, which makes the gradient unstable. We therefore consider the steepest descent direction.

We start by showing the general steepest descent direction.

$$\mathbf{d}^* = \underset{\mathbf{d}\in\mathbb{R}^D}{\mathrm{argmin}}\{\langle\nabla_{\boldsymbol{\mu}}\phi,\mathbf{d}\rangle \mid \|\mathbf{d}\|_M = 1\} \qquad \langle\mathbf{d},M\mathbf{d}\rangle = 1 \Rightarrow M = A^\mathsf{T}A \tag{29}$$

$$= A^{-1}\underset{\mathbf{x}\in\mathbb{R}^D}{\mathrm{argmin}}\{\langle\nabla_{\boldsymbol{\mu}}\phi,A^{-1}\mathbf{x}\rangle \mid \|\mathbf{x}\|_2 = 1\} \qquad \langle A\mathbf{d},A\mathbf{d}\rangle = 1 \Rightarrow \mathbf{x} = A\mathbf{d} \tag{30}$$

$$= A^{-1}\underset{\mathbf{x}\in\mathbb{R}^D}{\mathrm{argmin}}\{\langle A^{-\mathsf{T}}\nabla_{\boldsymbol{\mu}}\phi,\mathbf{x}\rangle \mid \|\mathbf{x}\|_2 = 1\}. \qquad \langle\mathbf{x},\mathbf{x}\rangle = 1 \text{ and } \mathbf{d} = A^{-1}\mathbf{x} \tag{31}$$

$$\tag{32}$$

Using the Cauchy-Schwarz inequality $(-\|\mathbf{x}\|_2\|\mathbf{y}\|_2 \leq \langle\mathbf{x},\mathbf{y}\rangle)$ for the optimization problem (31), we get that the minimizer is equal to

$$-\left\|A^{-\mathsf{T}}\nabla_{\boldsymbol{\mu}}\phi\right\|_2\|\mathbf{x}\|_2 \leq \langle A^{-\mathsf{T}}\nabla_{\boldsymbol{\mu}}\phi,\mathbf{x}\rangle \Rightarrow \mathbf{x}^* = -\frac{A^{-\mathsf{T}}\nabla_{\boldsymbol{\mu}}\phi}{\|A^{-\mathsf{T}}\nabla_{\boldsymbol{\mu}}\phi\|_2}, \tag{33}$$

and thus, by plugging the result of (33) in to (31), we get that the steepest descent direction is

$$\mathbf{d}^* = -\frac{A^{-1}A^{-\mathsf{T}}\nabla_{\boldsymbol{\mu}}\phi}{\|A^{-1}A^{-\mathsf{T}}\nabla_{\boldsymbol{\mu}}\phi\|_2} = -\frac{(A^\mathsf{T}A)^{-1}\nabla_{\boldsymbol{\mu}}\phi}{\sqrt{\langle A^{-\mathsf{T}}\nabla_{\boldsymbol{\mu}}\phi,\langle A^{-\mathsf{T}}\nabla_{\boldsymbol{\mu}}\phi,\rangle\rangle}} = -\frac{M^{-1}\nabla_{\boldsymbol{\mu}}\phi}{\sqrt{\langle\nabla_{\boldsymbol{\mu}}\phi,M^{-1}\nabla_{\boldsymbol{\mu}}\phi\rangle}}$$

$$\Rightarrow \mathbf{d}^* = -\frac{M^{-1}\nabla_{\boldsymbol{\mu}}\phi}{\|M^{-1}\nabla_{\boldsymbol{\mu}}\phi\|_M}. \tag{34}$$

In our case, the $M = \boldsymbol{\Sigma}^{-1}$, and thus, we get that the steepest descent direction of the objective function of the LAND model is

$$\mathbf{d}^* = \frac{1}{N}\sum_{n=1}^{N}\mathrm{Log}_{\boldsymbol{\mu}}(\mathbf{x}_n) - \frac{\mathcal{Z}}{\mathcal{C}(\boldsymbol{\mu},\boldsymbol{\Sigma})\cdot S}\sum_{s=1}^{S}m(\boldsymbol{\mu},\mathbf{v}_s)\mathbf{v}_s, \tag{35}$$

where we omit the denominator $\|\nabla_{\boldsymbol{\mu}}\phi(\boldsymbol{\mu},\boldsymbol{\Sigma})\|_2$, since this is just a scaling factor, which will be captured by the stepsize. This avoid problems that appears due to large condition numbers of $\boldsymbol{\Sigma}$.

# 3 Gradient Direction for the Covariance

We decompose the $\boldsymbol{\Sigma}^{-1} = \mathbf{A}^{\mathsf{T}}\mathbf{A}$. In addition, we rewrite the inner product as follows

$$\langle \mathrm{Log}_{\boldsymbol{\mu}}(\mathbf{x}_n), \mathbf{A}^{\mathsf{T}}\mathbf{A}\mathrm{Log}_{\boldsymbol{\mu}}(\mathbf{x}_n)\rangle = tr(\mathrm{Log}_{\boldsymbol{\mu}}(\mathbf{x}_n)^{\mathsf{T}}\mathbf{A}^{\mathsf{T}}\mathbf{A}\mathrm{Log}_{\boldsymbol{\mu}}(\mathbf{x}_n)) \tag{36}$$

$$= tr(\mathbf{A}\mathrm{Log}_{\boldsymbol{\mu}}(\mathbf{x}_n)\mathrm{Log}_{\boldsymbol{\mu}}(\mathbf{x}_n)^{\mathsf{T}}\mathbf{A}^{\mathsf{T}}) \tag{37}$$

$$\Rightarrow \frac{\partial}{\partial \mathbf{A}}[tr(\mathbf{A}\mathrm{Log}_{\boldsymbol{\mu}}(\mathbf{x}_n)\mathrm{Log}_{\boldsymbol{\mu}}(\mathbf{x}_n)^{\mathsf{T}}\mathbf{A}^{\mathsf{T}})] = 2\mathbf{A}\mathrm{Log}_{\boldsymbol{\mu}}(\mathbf{x}_n)\mathrm{Log}_{\boldsymbol{\mu}}(\mathbf{x}_n)^{\mathsf{T}}, \tag{38}$$

where $tr(\cdot)$ is the trace operator. Then the gradient of the objective with respect the matrix $\mathbf{A}$ is

$$\nabla_{\mathbf{A}}\phi(\boldsymbol{\mu},\boldsymbol{\Sigma}) = \frac{\partial}{\partial \mathbf{A}}\left[\frac{1}{2N}\sum_{n=1}^{N}\langle \mathrm{Log}_{\boldsymbol{\mu}}(\mathbf{x}_n), \mathbf{A}^{\mathsf{T}}\mathbf{A}\mathrm{Log}_{\boldsymbol{\mu}}(\mathbf{x}_n)\rangle + \log(\mathcal{C}(\boldsymbol{\mu},\boldsymbol{\Sigma}))\right] \tag{39}$$

$$= \frac{1}{2N}2\mathbf{A}\sum_{n=1}^{N}\mathrm{Log}_{\boldsymbol{\mu}}(\mathbf{x}_n)\mathrm{Log}_{\boldsymbol{\mu}}(\mathbf{x}_n)^{\mathsf{T}} \tag{40}$$

$$+ \frac{1}{\mathcal{C}(\boldsymbol{\mu},\boldsymbol{\Sigma})}\int_{\mathcal{M}}\frac{\partial}{\partial \mathbf{A}}\left[\exp\left(-\frac{1}{2}\langle \mathrm{Log}_{\boldsymbol{\mu}}(\mathbf{x}), \mathbf{A}^{\mathsf{T}}\mathbf{A}\mathrm{Log}_{\boldsymbol{\mu}}(\mathbf{x})\rangle\right)\right]\mathrm{d}\mathcal{M}(\mathbf{x}) \tag{41}$$

$$= \frac{1}{N}\mathbf{A}\sum_{n=1}^{N}\mathrm{Log}_{\boldsymbol{\mu}}(\mathbf{x}_n)\mathrm{Log}_{\boldsymbol{\mu}}(\mathbf{x}_n)^{\mathsf{T}} \tag{42}$$

$$- \frac{\mathbf{A}}{\mathcal{C}(\boldsymbol{\mu},\boldsymbol{\Sigma})}\int_{\mathcal{M}}\mathrm{Log}_{\boldsymbol{\mu}}(\mathbf{x})\mathrm{Log}_{\boldsymbol{\mu}}(\mathbf{x})^{\mathsf{T}}\exp\left(-\frac{1}{2}\langle \mathrm{Log}_{\boldsymbol{\mu}}(\mathbf{x}), \mathbf{A}^{\mathsf{T}}\mathbf{A}\mathrm{Log}_{\boldsymbol{\mu}}(\mathbf{x})\rangle\right)\mathrm{d}\mathcal{M}(\mathbf{x}) \tag{43}$$

$$= \frac{1}{N}\mathbf{A}\sum_{n=1}^{N}\mathrm{Log}_{\boldsymbol{\mu}}(\mathbf{x}_n)\mathrm{Log}_{\boldsymbol{\mu}}(\mathbf{x}_n)^{\mathsf{T}} \tag{44}$$

$$- \frac{\mathbf{A}}{\mathcal{C}(\boldsymbol{\mu},\boldsymbol{\Sigma})}\int_{\mathcal{T}_{\boldsymbol{\mu}}\mathcal{M}}m(\boldsymbol{\mu},\mathbf{v})\mathbf{v}\mathbf{v}^{\mathsf{T}}\exp\left(-\frac{1}{2}\langle \mathbf{v}, \boldsymbol{\Sigma}^{-1}\mathbf{v}\rangle\right)\mathrm{d}\mathbf{v}. \tag{45}$$

Finally, treating the integral as an expectation problem and using Monte Carlo integration, we get that the gradient is

$$\nabla_{\mathbf{A}}\phi(\boldsymbol{\mu},\boldsymbol{\Sigma}) = \mathbf{A}\left[\frac{1}{N}\sum_{n=1}^{N}\mathrm{Log}_{\boldsymbol{\mu}}(\mathbf{x}_n)\mathrm{Log}_{\boldsymbol{\mu}}(\mathbf{x}_n)^{\mathsf{T}} - \frac{\mathcal{Z}}{\mathcal{C}(\boldsymbol{\mu},\boldsymbol{\Sigma})\cdot S}\sum_{s=1}^{S}m(\boldsymbol{\mu},\mathbf{v}_s)\mathbf{v}_s\mathbf{v}_s^{\mathsf{T}}\right]. \tag{46}$$

# 4 Gradients for the LAND Mixture Model

Similarly the LAND mixture model are

$$\nabla_{\boldsymbol{\mu}_k}\psi(\boldsymbol{\Theta}) = -\boldsymbol{\Sigma}_k^{-1}\left[\sum_{n=1}^{N}r_{nk}\mathrm{Log}_{\boldsymbol{\mu}_k}(\mathbf{x}_n) - \frac{\mathcal{Z}\cdot R_k}{\mathcal{C}_k(\boldsymbol{\mu}_k,\boldsymbol{\Sigma}_k)\cdot S}\sum_{s=1}^{S}m(\boldsymbol{\mu}_k,\mathbf{v}_s)\mathbf{v}_s\right] \tag{47}$$

$$\nabla_{\mathbf{A}_k}\psi(\boldsymbol{\Theta}) = \mathbf{A}_k\left[\sum_{n=1}^{N}r_{nk}\mathrm{Log}_{\boldsymbol{\mu}_k}(\mathbf{x}_n)\mathrm{Log}_{\boldsymbol{\mu}_k}(\mathbf{x}_n)^{\mathsf{T}} - \frac{\mathcal{Z}\cdot R_k}{\mathcal{C}_k(\boldsymbol{\mu}_k,\boldsymbol{\Sigma}_k)\cdot S}\sum_{s=1}^{S}m(\boldsymbol{\mu}_k,\mathbf{v}_s)\mathbf{v}_s\mathbf{v}_s^{\mathsf{T}}\right] \tag{48}$$

where $R_k = \sum_{n=1}^{N}r_{nk}$, and the responsibilities $r_{nk} = \frac{\pi_k p_{\mathcal{M}}(\mathbf{x}_n \mid \boldsymbol{\mu}_k,\boldsymbol{\Sigma}_k)}{\sum_{l=1}^{K}\pi_l p_{\mathcal{M}}(\mathbf{x}_n \mid \boldsymbol{\mu}_l,\boldsymbol{\Sigma}_l)}$.

# 5 Algorithms

In this section we present the algorithms for: 1) estimating the normalization constant $\mathcal{C}(\boldsymbol{\mu}, \boldsymbol{\Sigma})$, 2) maximum likelihood estimation of the LAND, and 3) fitting the LAND mixture model.

---
**Algorithm 1** The estimation of the normalization constant $\mathcal{C}(\boldsymbol{\mu}, \boldsymbol{\Sigma})$

---
**Input:** the given data $\{\mathbf{x}_n\}_{n=1}^N$, the $\boldsymbol{\mu}$, $\boldsymbol{\Sigma}$, the number of samples $S$
**Output:** the estimated $\hat{\mathcal{C}}(\boldsymbol{\mu}, \boldsymbol{\Sigma})$
  1: sample $S$ tangent vectors $\mathbf{v}_s \sim \mathcal{N}(0, \boldsymbol{\Sigma})$ on $\mathcal{T}_{\boldsymbol{\mu}}\mathcal{M}$
  2: map the $\mathbf{v}_s$ on $\mathcal{M}$ as $\mathbf{x}_s = \text{Exp}_{\boldsymbol{\mu}}(\mathbf{v}_s), \ s = 1, \dots, S$
  3: compute the normalization constant $\hat{\mathcal{C}}(\boldsymbol{\mu}, \boldsymbol{\Sigma}) = \frac{\mathcal{Z}}{S} \sum_{s=1}^S \sqrt{|\mathbf{M}(\mathbf{x}_s)|}$

---

---
**Algorithm 2** LAND maximum likelihood

---
**Input:** the data $\{\mathbf{x}_n\}_{n=1}^N$, stepsize $\alpha_{\boldsymbol{\mu}}, \alpha_A$, tolerance $\epsilon$
**Output:** the estimated $\hat{\boldsymbol{\mu}}, \ \hat{\boldsymbol{\Sigma}}, \ \hat{\mathcal{C}}(\hat{\boldsymbol{\mu}}, \hat{\boldsymbol{\Sigma}})$
  1: initialize $\boldsymbol{\mu}^0, \boldsymbol{\Sigma}^0$ and $t \leftarrow 0$
  2: **repeat**
  3:      estimate $\mathcal{C}(\boldsymbol{\mu}^t, \boldsymbol{\Sigma}^t)$ using Eq. 16
  4:      compute $d_{\boldsymbol{\mu}}\phi(\boldsymbol{\mu}^t, \boldsymbol{\Sigma}^t)$ using Eq. 35
  5:      $\boldsymbol{\mu}^{t+1} \leftarrow \text{Exp}_{\boldsymbol{\mu}^t}(\alpha_{\boldsymbol{\mu}} d_{\boldsymbol{\mu}}\phi(\boldsymbol{\mu}^t, \boldsymbol{\Sigma}^t))$
  6:      estimate $\mathcal{C}(\boldsymbol{\mu}^{t+1}, \boldsymbol{\Sigma}^t)$ using Eq. 16
  7:      compute $\nabla_{\mathbf{A}}\phi(\boldsymbol{\mu}^{t+1}, \boldsymbol{\Sigma}^t)$ using Eq. 46
  8:      $\mathbf{A}^{t+1} \leftarrow \mathbf{A} - \alpha_{\mathbf{A}} \nabla_{\mathbf{A}}\phi(\boldsymbol{\mu}^{t+1}, \boldsymbol{\Sigma}^t)$
  9:      $\boldsymbol{\Sigma}^{t+1} \leftarrow [(\mathbf{A}^{t+1})^\intercal \mathbf{A}^{t+1}]^{-1}$
10:      $t \leftarrow t + 1$
11: **until** $\left\| \phi(\boldsymbol{\mu}^{t+1}, \boldsymbol{\Sigma}^{t+1}) - \phi(\boldsymbol{\mu}^t, \boldsymbol{\Sigma}^t) \right\|_2^2 \leq \epsilon$

---

---
**Algorithm 3** LAND mixture model

---
**Input:** the data $\{\mathbf{x}_n\}_{n=1}^N$, $\{\alpha_{\boldsymbol{\mu}_k}, \alpha_{\mathbf{A}_k}\}_{k=1}^K$, tolerance $\epsilon$
**Output:** the estimated $\{\hat{\boldsymbol{\mu}}_k, \ \hat{\boldsymbol{\Sigma}}_k, \ \hat{\mathcal{C}}_k, \ \hat{\pi}_k\}_{k=1}^K$
  1: initialize the $\{\boldsymbol{\mu}_k^0, \ \boldsymbol{\Sigma}_k^0, \ \mathcal{C}_k^0, \ \pi_k^0\}_{k=1}^K$ and $t \leftarrow 0$
  2: **repeat**
  3:      **Expectation step**:
  4:      compute the responsibilities $r_{nk} = \frac{\pi_k p_{\mathcal{M}}(\mathbf{x}_n \mid \boldsymbol{\mu}_k, \boldsymbol{\Sigma}_k)}{\sum_{t=1}^K \pi_t p_{\mathcal{M}}(\mathbf{x}_n \mid \boldsymbol{\mu}_t, \boldsymbol{\Sigma}_t)}$
  5:      **Maximization step**:
  6:      **for** $k = 1, \dots, K$ **do**
  7:          estimate $\mathcal{C}_k(\boldsymbol{\mu}_k^t, \boldsymbol{\Sigma}_k^t)$ using Eq. 16
  8:          compute from Eq. 47 the $d_{\boldsymbol{\mu}}\phi(\boldsymbol{\mu}_k^t, \boldsymbol{\Sigma}_k^t)$
  9:          $\boldsymbol{\mu}_k^{t+1} \leftarrow \text{Exp}_{\boldsymbol{\mu}_k^t}(\alpha_{\boldsymbol{\mu}_k} d_{\boldsymbol{\mu}}\phi(\boldsymbol{\mu}_k^t, \boldsymbol{\Sigma}_k^t))$
10:          estimate $\mathcal{C}_k(\boldsymbol{\mu}_k^t, \boldsymbol{\Sigma}_k^t)$ using Eq. 16
11:          compute from Eq. 48 the $\nabla_{\mathbf{A}_k}\phi(\boldsymbol{\mu}_k^{t+1}, \boldsymbol{\Sigma}_k^t)$
12:          $\mathbf{A}_k^{t+1} \leftarrow \mathbf{A}_k^t - \alpha_{\mathbf{A}_k} \nabla_{\mathbf{A}}\phi(\boldsymbol{\mu}_k^{t+1}, \boldsymbol{\Sigma}_k^t)$
13:          $\boldsymbol{\Sigma}_k^{t+1} \leftarrow [(\mathbf{A}_k^{t+1})^\intercal \mathbf{A}_k^{t+1}]^{-1}$
14:          $\pi_k = \frac{1}{N} \sum_{n=1}^N r_{nk}$
15:      **end for**
16:      $t \leftarrow t + 1$
17: **until** $\left\| \psi(\boldsymbol{\Theta}^{t+1}) - \psi(\boldsymbol{\Theta}^t) \right\|_2^2 \leq \epsilon$

---

## 5.1 Stepsize Selection

The LAND objective is expensive to evaluate due to the dependency on $\text{Log}_\mu(\mathbf{x}_n)$. This imply that a line-search is infeasible for selecting a stepsize. Thus, we use the following common trick. Each stepsize is given in the start of the algorithm. If the objective increased after an update, we reduce the corresponding stepsize as $\alpha = 0.75 \cdot \alpha$, and if the objective reduced, then $\alpha = 1.1 \cdot \alpha$.

## 5.2 Initialization Issues

The initialization of the LAND is important, as well as for the mixture model. We discuss two different initializations plus one specifically for the mixture model.

1. **Random**: we initialize the LAND mean with a random point on the manifold. The initial covariance is the empirical covariance of the tangent vectors. This initialization can be used also for the mixture model, with $K$ random starting points. Then, we cluster the points and the covariances are initialized using empirical estimators.

2. **Least Squares**: we initialize the LAND with the intrinsic least squares mean, and the covariance with the empirical estimator. This initialization can be used also for the mixture model, using the extension of the $k$-means on Riemannian manifolds, and then, the points of each cluster for the empirical covariances.

3. **GMM**: we initialize the LAND mixture model centres with the result of the GMM. For the empirical covariances, we use the points that belong to each cluster from the GMM solution.

## 5.3 Stopping criterion

Our objective function is non-convex, thus, as stopping criterion we use the change of the objective value. In particular, we stop the optimization when $\left\| \phi(\boldsymbol{\mu}^{t+1}, \boldsymbol{\Sigma}^{t+1}) - \phi(\boldsymbol{\mu}^t, \boldsymbol{\Sigma}^t) \right\|_2^2 \leq \epsilon$, for some $\epsilon$ given by the user. The same stopping criterion is used for the mixture model.

# 6 Experiments

In this section we provide additional illustrative experiments.

## 6.1 Estimating the Normalization Constant

In order to show the consistency of the normalization constant estimation with respect to the number of samples, we conduct the following experiment. We used the data from the first synthetic experiment of the paper. Then for a grid $100 \times 100$ on the $\mathcal{T}_\mu \mathcal{M}$ we computed the corresponding $m(\boldsymbol{\mu}, \mathbf{v})$ values. Thus, we computed the numerical integral on the tangent space using trapezoidal numerical integration. Then we estimated the normalization constant using our approach for sample sizes $S = 100 : 100 : 3000$ and for 10 different runs. From the result in Fig. 1 we observe that the numerical scheme we provide, approximates well the normalization constant that we computed numerically.

## 6.2 MNIST digit 1 data

In this experiment we used the digit 1 from the MNIST dataset. We sample 200 points and using PCA we projected them onto the first 2 principal components. Then we fitted LAND, a least squares model, and a normal distribution. From the result Fig. 2 we observe that the LAND model approximates efficiently the underlying distribution of the data. Also, the least squares model has a similar performance, since it takes under consideration the underlying manifold. However, is obvious that it overfits the given data, and gives significant probability to low density areas. On the other hand, the linear model has poor performance, due to the linear distance measure.

Figure 1: The normalization constant estimation for different sample sizes $S$. The black line denotes the trapezoidal numerical integral, and the dashed red line the mean value of the estimators using our proposed method.

Figure 2: The MNIST digit 1 projected onto the 2 first principal components experiment. *Left*: the LAND model approximates efficiently the data distribution. *Center*: the least squares model approximates the distribution, but it overfits the given data. *Right*: the normal distribution has poor performance due to the linear distance measure.

## 6.3 The Sleep Stages Experiment

Here we present the feature extraction result for 3 factors. From the Fig. 3 we observe that actually, the derived data have a manifold structure. Moreover, we see that the characteristics of the data i.e., the EEG measurement, varies a lot between the 3 subjects.

Figure 3: *Top row*: The given data, after the feature extraction procedure for 3 factors for three subjects. *Bottom row*: the F-measure for different values of $\sigma$ (subject "s151").

## 6.4 The Clustering Problem for the Synthetic Data

Due to space limitations, we were not able to present the result of the least squares mixture model for the clustering problem of the synthetic data, thus, we present here the result. From the Fig. 4 we observe that indeed the LAND can approximates efficiently the underlying distributions of the clusters. Even thought the least squares mixture model takes under consideration the underlying structure of the data, it fails to reveal precisely the distributions of the clusters. Thus, we argue that our maximum likelihood estimates are better than the least squares estimates. On the other hand, the GMM fails even to find the correct means of the distributions.

Figure 4: The clustering problem for two synthetic datasets. *Left*: the LAND mixture model approximates efficiently the underlying distributions of the clusters. *Center*: the least squares fails to reveal precisely the distributions of the clusters. *Right*: the GMM due to the linear distance measure fails even to find the correct means of the distributions.

## 6.5 The Contour Plots for the Synthetic Data

Additionally to the results presented in the main paper, in Fig. 5 we present the contours of all the fitted models and for all the numbers of components, where the advantages of the LAND are obvious. Especially, when $K = 1$ we observe that the LAND approximates well the underlying distribution, while even though the least squares estimator reveals the nonlinearity of the distribution, as we discussed in the paper the covariance overfits the given data.

Furthermore, when $K$ increases the LAND components locally become almost linear Gaussians, since the geodesics will almost be straight lines. However, even in this case the LAND mixture model is more flexible than the Gaussian mixture model, see the result for $K = 4$. Also, the LAND does not overfit the given data, as the least squares mixture model does, since the probability mass is more concentrated around the means, see the result for $K = 2$.

## 6.6 Motion Capture Data

We conducted an experiment using motion capture data from *CMU Motion Capture Database*[1]. Specifically, we picked two movements motion: 16 from subject 22 (jumping jag), and the subject 9 (run). Each data point corresponds to a human pose. We projected the data onto the first 2 and 3 principal and we fitted a LAND mixture model and a Gaussian mixture model for $K = 2$. From the results in Fig. 6 we see that the LAND means fall inside the data, while the GMM means are actually outside of the manifold.

Figure 5: Synthetic data and the fitted models. From top to bottom we present the results for $K = 1, 2, 3, 4$, respectively. *Left*: the contours of the LAND mixture model. *Center*: the contours of the least squares mixture model. *Right*: the contours of the Gaussian mixture model.

Figure 6: Motion capture experiment. *Left*: the experiment for the run sequence. *Right*: the experiment for the jumping jag sequence.

## 6.7 Scalability of Geodesic Computations

A scalability concern is that the underlying ODEs are computationally more demanding in high dimensions, and more specifically, we are interested in the logarithm map. We conducted a supplementary experiment on the MNIST data, reporting the ODE solver running time as a function of input dimensionality. In particular, we fix a point and we compute the running time of the logarithm map between this point and 20 random chosen points, for a set of the dimensions of the feature space. From the result in Fig. 7 we observe that the current implementation scales to approximately 50 dimensions, where it becomes impractical.

Figure 7: Scalability experiment.

## 6.8 Model Selection

We used the standard AIC and BIC criteria,

$$BIC = -2 \cdot \ln(L) + \nu \cdot \ln(N) \tag{49}$$
$$AIC = -2 \cdot \ln(L) + 2 \cdot \nu \tag{50}$$

where $L \in \mathbb{R}$ is the log-likelihood of the model, and $\nu \in \mathbb{R}$ is the number of free parameters. The optimal number of components $K$ can then be chosen to minimize either criteria. Note that the LAND and the GMM are not normalized under the same measure, so their likelihoods are not directly comparable. However, we can select the optimal $K$ for each method separately.

We used the synthetic data from the first experiment in the paper. From the results in Fig. 8 we observe that the optimal LAND model is achieved for $K = 1$, while the for the least squares estimators and the GMM, the optimal is achieved for $K = 3$ and $K = 4$ respectively. Thus, we argue that the less complex LAND model with only one component, is able to reveal the underlying distribution, while the other two methods need more components resulting to more complex models.

Figure 8: Model selection experiment. *Left*: the AIC criterion. *Right*: the BIC criterion.

## Footnotes

[1]http://mocap.cs.cmu.edu/