[Reviews · NeurIPS 2016]

Reviewer 1

Summary

The paper proposes LAND, a new parametric metric defined over a nonparametric space, by bridging concepts from metric learning/manifold learning and Riemannian statistics. The parametric metric learned is a Riemannian normal distribution as defined in Pennec [15]. The authors propose a maximum likelihood algorithm to estimate the mean and covariance of the Riemannian normal distribution, i.e. a Mahalanobis distance for Riemannian manifolds. Distances between points are defined as geodesics along the Riemannian manifold as in Hauberg et al. [9], and they rely on learning a local metric. Here, the authors propose to learn the metric by using the inverses of the local (diagonal) covariances. The authors also consider mixtures of LANDs.

Qualitative Assessment

The paper brings new insights into the connection between metric learning and Riemannian statistics. Having parametric and generative models that can go beyond the training data is very useful in many scenarios. However, what I am mainly concerned is its applicability to real world data which is both large scale and high-dimensional. LAND requires the computation of a (here, diagonal) covariance matrix for each data point and of its inverse, which in high dimensions are known to be hard problems. The authors do mention high-dimensionality as a possible issue (lines 265-266), but the paper would need an analysis of the complexity and limitations of the metric introduced. How high is here high-dimensional? 3, 5, 10, 100 dimensions? The literature on metric learning is fairly vast and I find that the paper needs to be better fitted in this context for future readers. Hauberg et al. [9] already discuss some of the literature on metric learning, e.g., Frome et al. [1,2], Malisiewicz and Efros [6], who all learn a diagonal metric tensor for every point similar to LAND. Local covariances have also been used in methods like local PCA (Kambhatla – Dimension reduction by local principal component analysis) even if not for every point. Lines 101-103: For learning the local metric, the authors use a Gaussian kernel to define locality. How does the parameter sigma influence the results? The choice of sigma is only discussed for the sleep data in Sect. 4.2., but not for the synthetic data in Sect. 4.1. The paper would benefit from robustness tests for sigma. Is sigma the same for all points, whether in low or high-density regions? Fig. 4 compares LAND also against intrinsic estimators and the results are fairly similar, but the authors never show visual results with these intrinsic estimators in Figs. 5 and 6. Instead, visualization results are shown against GMM which are already shown in Fig. 4 to perform badly. This is therefore less insightful. Some visual results using the intrinsic model are shown in the supplementary material (Fig. 2 and 4) and they outperform GMM. Therefore, these results should probably replace the GMM results in the initial paper or make room for both. It would be interesting to do an analysis of the computational complexity of LAND vs. intrinsic estimators to see which method should be favored over the other, and for which data structures. The authors also consider mixtures of LANDs. I think this is one of the most interesting applications of LAND, as data is rarely generated by one component. This aspect would be worth expanding. The paper is very well written, it is a pleasure to read.

Confidence in this Review

2-Confident (read it all; understood it all reasonably well)


Reviewer 2

Summary

This brief paper extends the common multivariate normal density by replacing the Euclidean metric with a locally adaptive Riemannian metric. The result is a generalization of the normal distribution to the manifold learning setting with wide applicability (as illustrated by the reported experients)

Qualitative Assessment

This brief paper extends the common multivariate normal density by replacing the Euclidean metric with a locally adaptive Riemannian metric. The result is a generalization of the normal distribution to the manifold learning setting with wide applicability (as illustrated by the reported experients) Importantly, it provides, very sinthetically, a complete account of a parameter estimation procedure of the maximum likelihood type. The extension to a mixture of LANDS makes it particularly practical as a 'competitor' to the now standard GMM. The experiments are (at least for this reviewer) quite exciting and open a wide avenue for future research. Language, estructure, bibliography and relevance to the NIPS remit are excellent.

Confidence in this Review

2-Confident (read it all; understood it all reasonably well)


Reviewer 3

Summary

This paper proposes to model the observation space with a hand-crafted Riemannian metric, and define a curved normal distribution based on the associated Riemannian geometry, so that the data manifold can be better captured.

Qualitative Assessment

The paper addresses a fundamental problem on how to define a Riemannian metric based on a set of high-dimensional real-valued observations, and then how to define and estimate a curved Gaussian distribution based on this geometry. This presents a useful tool and an interesting application of Riemannian geometry in statistical learning. The first part of the proposed method defines the underlying Riemannian geometry in section 3.1. The title suggests that it "learns a metric". However the metric is actually hand-crafted in equation 7 based on two hyper-parameters rho and sigma. These hyper-parameters are fixed empirically in advance. I am guessing that sigma is a sensitive parameter. The paper should discuss in more detail what is the intuitive effect when varying sigma, and how to select this sigma. With this part missing the paper is incomplete. Is it possible to merge the metric learning stage with the LAND learning stage, so that rho and sigma can be learned? A beauty of the geometric approach is to be invariant to coordinate transformations. However the metric in equation 7 is not invariant. For example, if the data forms a line in a 2D observation space. We will have completely different metrics in a coordinate system with its x-axis parallel to this line, and in a coordinate system that forms an angle, e.g. 45 degrees, to this line. It is useful to discuss this in more detail, including whether the corresponding full (rather than diagonal) metric is invariant, and give more intuitions of the metric in equation 7. There is a rich literature in metric learning and manifold learning with connection to Riemannian geometry that can be related to this work (see google "Riemannian" "metric learning" "manifold learning"). As a main criticism, in the first experiment, the comparison is not so meaningful because that obviously the more flexible LAND can describe the data better with a smaller number of component. However GMM can describe the data better with a large number of components when LAND will overfit. The author(s) are suggested to redesign the experiment, to let each compared method to choose its favourable number of components based on certain criterion (e.g. BIC), then compare the corresponding performance. LAND is still expected to perform better because it describe better the manifold structure, and this comparison is more meaningful. It is nice to see how LAND with a large number of components will overfit. LAND is obviously a more expensive procedure than estimating a Gaussian. Is there any quantitative measurements or experiments to make it clear how expensive it is? What is the limit of the scale that LAND can handle? Is it possible to learn LAND on MNIST and compare it with Gaussian mixture learning? ====== After reading the authors' response LAND is a more flexible model than Gaussian distribution. If you increase the number of components, each local LAND will be more vulnerable to local noises as compared to a local Gaussian. "The geodesics will almost be straight lines", true, but it is still curves, and these curves will lead to overfiting. I highly suggest that the authors to revise this experiment in the revision.

Confidence in this Review

2-Confident (read it all; understood it all reasonably well)


Reviewer 4

Summary

The paper proposes an alternative for the Gaussian Mixture Models, namely a Locally Adaptive Normal Distribution. The proposed distribution is based on Riemannian Manifolds. The paper proposes an MLE estimator for the parameters of the distribution, and evaluates the method using experiments on both artificial and real world data.

Qualitative Assessment

The paper is well written, with sound theory and good experiments. However, compared with Simo-Serra & al.[19], the paper does not provide a potent novelty. The differences between the proposed method and the state-of-the-art is not well explained before the reader reaches the conclusion. These differences should have been made clear since the introduction (Note that ref [19] is absent from the introduction).

Confidence in this Review

2-Confident (read it all; understood it all reasonably well)


Reviewer 5

Summary

Summary: the paper proposes a locally adaptive normal distribution based on a geodesic manifold traversal distance, learned adaptively from the data. The distance replaces the standard Mahalanobis distance used in the Gaussian distribution. The extension to mixture of LANDs is also provided. The performance is evaluated on a synthetic dataset as well as a real-world dataset of EEG measurements.

Qualitative Assessment

Major comments: the proposed idea of using a gedesic distance, similar to Isomaps, is very interesting. However, there are few weak spots in the clarity of the paper. First, the definition of the inverse of the metric tensor in Eq. (7) seems ad hoc. Can authors provide more intution on the idea of using a Gaussian kernel for the weights in a weighted sum of local covariances? Also, I got lost between the definition of the local covarance matrix in Eq. (7) and later, the MAP estimation of the same quantity in Eq. (11). Do these refer to the same quantity? Next, the experimental evaluations are rather poor. But my main concern is the scalibility and computational efficiency of the method. For example, what is computational compleixity and the running time compared to standard EM? In my opinion, the paper has an interesting theoretical contribution, but might not be very applicable in practice. Minor comments: Algorithm 1, lines 4 and 5, d -> \nabla

Confidence in this Review

1-Less confident (might not have understood significant parts)


Reviewer 6

Summary

In this paper the authors present a locally adaptive normal distribution. Instead of measuring distances via a Euclidean distance they define a Riemannian metric M(x) which varies smoothly across the data set. Given the R. Metric M(x) geodesic distances along the manifold can be calculated via the exponential and logarithmic map. The authors use the Euler-Lagrange equations which define a set of ordinary differential equations for estimating the geodesic distances. The logarithm map takes points from the manifold (y) into the tangent plane (at x) such that the length of the resulting vector is the geodesic distance from x to y along the manifold. The authors then apply a normal distribution computing a Mahalanobis distance on points mapped by the logarithm map. This inner product is used to compute distances from a point x to the mean instead of Euclidean distances. The authors then derive estimators for the mean and covariance via maximum likelihood. Since the solutions are not available in closed form they present a steepest descent algorithm for estimating them. Finally, since the normalization constant is also not available in closed form the authors present a Monte-Carlo approximation method. They apply this method to synthetic data generated on a manifold as well as in a clustering task. They compare to the standard mixture of Gaussians.

Qualitative Assessment

Overall I feel that the paper provides an interesting and useful extension of the normal distribution which adapts to local manifold structure. The authors combine work in metric estimation, computing geodesic distances, and learning distributions. I believe this work is worthy of publication. I have a few comments on my scores for the above: Technical quality Comments: The manifold techniques (using Euler-Lagrange equations, estimating logarithm maps) are sound and well explained. The Monte-Carlo estimation of the normalization constant and steepest descent algorithms are clever and well described. Analysis of the applications is lacking (sigma parameter appears to be picked by optimizing on training data?). Explanation of the usefulness of the learned distribution is also somewhat lacking. Novelty Comments: It appears the main contribution of this paper is combining previous work (using Kernels for a metric M(x), estimating geodesic distances via Euler-Lagrange, applying a normal distribution) and providing an algorithm for estimating the parameters (mean, covariance, normalization constant). Potential Impact Comments: The authors provide a fair evaluation of the GMM and LAND in the synthetic data (looking at log-likelihood on data generated by the learned distributions) but it's not clear how they chose the sigma parameter. The authors show that the LAND fits well on data generated along a manifold but don't show how well it compares to a GMM when data is generated according to a single normal distribution or GMM. In the real life (clustering) data they appear to choose the sigma value for the LAND via optimizing the F measure and then present that same F measure. What they should have done (or if they did made this clearer) is provide the F measure on a held-out test set to be fair. It is, therefore, hard to see if there is or is not a problem with over-fitting. It also would have been nice to examine very high dimensional data. One potentially useful application that the authors present is that the LAND could be used as a generative model for manifold data, however, they make the rather unusual manifold assumption that the data X is observed in D dimensions *and* the manifold dimension is also D (that is, points on the tangent plane are D-dimensional and the Metric M(x) is full rank). In manifold learning we're usually interested in data that lie on a d-dimensional manifold embedded into a D-dimensional Euclidean space where D is much larger than d. Clarity Comments: There's one particular notation point I found confusing: the overloading of the Sigma notation. In particular they say M = Sigma^-1 but then estimate it via a (local) diagonal Kernel matrix. Is this Simga also the same as the (fitted) covariance matrix? If so, is the kernel just an initialization? This would be strange since M(x) should vary with x but the fitted covariance matrix is a single parameter. In the proof of the steepest decent they explicitly say say that "in our case M = Sigma^-1" and use that substitution. Clarification here would be nice. In Sec 4.2 they don't state but I assume cluster membership was chosen by picking 2 clusters and computing closest (geodesic?) distance to the cluster means. Overall I felt the manifold introduction was well explained (at least for someone familiar with Riemannian manifolds).

Confidence in this Review

2-Confident (read it all; understood it all reasonably well)